# An analytical approach for (U-Th)/He dating of goethite by sample encapsulation in quartz ampoules under vacuum, with an example from the Amerasian Basin, Arctic Ocean

Olga Valentinovna Yakubovich[1,2], Natalia Pavlovna Konstantinova[2,3], Maria Olegovna Anosova[4], Mary Markovna Podolskaya[5], Elena Valerevna Adamskaya[2]

[1]Department of geochemistry, Saint-Petersburg University, St. Petersburg, 199034, Russia
[2]Laboratory of Isotope Geology, Institute of Precambrian Geology and Geochronology RAS, St. Petersburg, 199034, Russia
[3]Institute for Geology and Mineral Resources of the Ocean (VNIIOkeangeologia), St. Petersburg, 190121, Russia
[4]Vernadskiy institute of Geochemistry and Analytical Chemistry, Moscow, 119991, Russia
[5]AnalyteMe, Reus, Tarragona, 43201, Spain

*Correspondence to*: Olga Yakubovich (olya.v.yakubovich@gmail.com)

**Abstract**

We propose an analytical approach for (U-Th)/He dating of Fe-(oxyhydr)oxides, that includes sealing samples in quartz ampoules and demonstrates its suitability as a reliable tool for the investigation of geological processes. The (U-Th)/He ages of goethite clasts and vein from Fe- and Mn-oxide cemented rocks recovered from the slope of Chukchi Borderland in the Amerasia Basin demonstrate reproducibility, yielding a weighted mean age of $8.6 \pm 0.3$ Ma (n=4) and $4.8 \pm 0.4$ Ma (n=2), respectively, providing insights into the Neogene mineralization history of the region. This study also focuses on the sample preparation technique, that might influence the (U-Th)/He ages. Our data indicate that some of U can be leached from the goethite during sonication by distilled water which might result in erroneous (U-Th)/He ages of multi-mineral grains. However, the analyzed goethite samples were formed at the specific underwater environment; so far it is not clear if the same behavior of U would be observed in a terrestrial supergene goethite.

## 1. Introduction

The (U-Th)/He dating method is based on the alpha-decay of U and Th that produce helium atoms. Traditionally, the $^4$He isotopic systems have been successfully applied to low-temperature thermochronology (Farley and Stockli, 2002). Recent developments in understanding how helium behaves in various minerals have extended the method applicability in geochronological studies (Yakubovich et al., 2019; Shukolyukov et al., 2012a; Flowers et al., 2023 and references therein).

Fe-oxides and Fe-hydroxides, including goethite (α-FeO(OH)), lepidocrocite (γ-FeO(OH)), hematite (α-Fe$_2$O$_3$), maghemite (γ-Fe$_2$O$_3$) and magnetite ( Fe$^{2+}$Fe$^{3+}$$_2$O$_4$), typically contain trace amounts of U and Th and therefore have been recognized as a potential geochronometer tool from the early days of geochronology (Strutt, 1908, 1909).

Goethite is one of the most common Fe- (oxy)hydroxide minerals formed during the hydrolyzation of rocks, making it a desirable mineral for dating various surface and subsurface geological processes. Helium diffusion studies ($^4$He/$^3$He spectra) revealed sufficient $^4$He retentivity in goethite an the range of the near-surface temperatures and make the mineral suitable for the (U-Th)/He weathering geochronology (Shuster et al., 2005). However, the accurate determination of He diffusion parameters is complicated by the dehydration of goethite during vacuum step-heating experiments (Farley et al., 2023).

The (U-Th)/He dating of goethite was applied successful in dating of weathering profiles (e.g.Monteiro et al., 2014; Riffel et al., 2016; Ansart et al., 2022), supergene ore formation (e.g.Vasconcelos et al., 2013; Heller et al., 2022; Verhaert et al., 2022), and diagenetic transformations (Reiners, 2014). The approach was also implemented successfully in dating deep-sea hydrothermal Fe-oxide mineralization (Benites et al., 2022). However, the dating of hydrogenetic Fe-Mn crusts is not robust due to the significant content of extraterrestrial He-rich dust and their high porosity, that prevent the accumulation of radiogenic He (Basu et al., 2006).

The (U-Th)/He dating of surface processes is challenging due to the multistage Fe-hydroxides formation. Several generations of the same phase intimately intergrow in a millions years time span (Vasconcelos et al., 2013; Monteiro et al., 2014; Heller et al., 2022). Presence of small inclusions of U- and Th-bearing contaminants may add difficulties to the interpretation of the isotopic results. Thus, high-resolution mineralogical and paragenetic characterization of the sample is required which typically includes optical observations accompanied by XRD, SEM and chemical analyses (e.g. Monteiro et al., 2014; Hofmann et al., 2017; Deng et al., 2017).

From the analytical point of view (U-Th)/He dating of goethite is challenging as well. The distribution of U and Th in the mineral is inhomogeneous (Shuster et al., 2005), therefore parental and daughter isotopes should be measured in the same sample. Helium release from the goethite must be carried out under strictly controlled laboratory heating conditions; otherwise, U and Th may be lost from the grains during He extraction rendering the results inaccurate (Vasconcelos et al., 2013). There are several approaches to overcome this issue such as heating in the presence of oxygen (Hofmann et al., 2020), using double-aliquot (Wernicke and Lippolt, 1993; Pidgeon et al., 2004), or multi-aliquot procedures (Wu et al., 2019). The last two require remarkably larger amount of material.

Here, we propose the alternative (U-Th)/He dating methodology using an example of goethite from the Chukchi Borderland, Arctic Ocean. The technique was originally developed for (U-Th)/He dating of native gold (Yakubovich et al. 2014) and pyrite (Yakubovich et al. 2020).

## 2. Samples

The Amerasia basin of the Arctic Ocean remains one of the Earth's least explored region (Brumley et al., 2015). The Chukchi Borderland and Mendeleev Ridge are known as Paleozoic continental blocks that occur within the Amerasia Arctic Ocean. During the U.S. and Russian research cruises fragments of Fe- and Mn-oxide mineralized rocks were collected from several sites of the northern Chukchi Borderland and central Mendeleev Ridge (Fig. 1A; Hein et al., 2017; Konstantinova et al., 2017). The subject of this study is the age dating of samples from dredge haul DR7 collected from 3400 m water depth (coordinates 65    78.53N, 156.68W).

The DR7 dredge haul consists of rock fragments that are extensively altered and finely sheared. Two different rock types were found. First one shows alternating yellow-brown and dark-brown layers, with dendrites of the dark-brown material in the yellow-brown laminae (Fig. 1B). Both layer types mainly comprise Fe-(hydro)oxides, but the dark-brown layers have a higher Mn-oxide content. Another rock type in DR7 is a breccia with poorly sorted predominantly angular to subangular clasts (Fig.
1C), that include pure Fe oxyhydroxide, basalt, and altered metasedimentary rocks. Mn- or Fe-oxide are found in some larger clasts. The breccia cement is composed predominately of dark brown Fe-oxyhydroxides with submetallic grey areas. The microstructure varies from bladed to nodular to massive. The breccia is predominantly cement-supported, indicating replacement during Fe- and Mn-oxide mineralization. Thus, these samples do not represent the widespread underwater hydrogenic Fe-Mn mineralization (Hein et al., 2000). Their morphology, structure, mineral and chemical composition
(especially high abundance of Fe-(hydro)oxides) indicate that they likely have a hydrothermal origin (Hein et al., 2024).

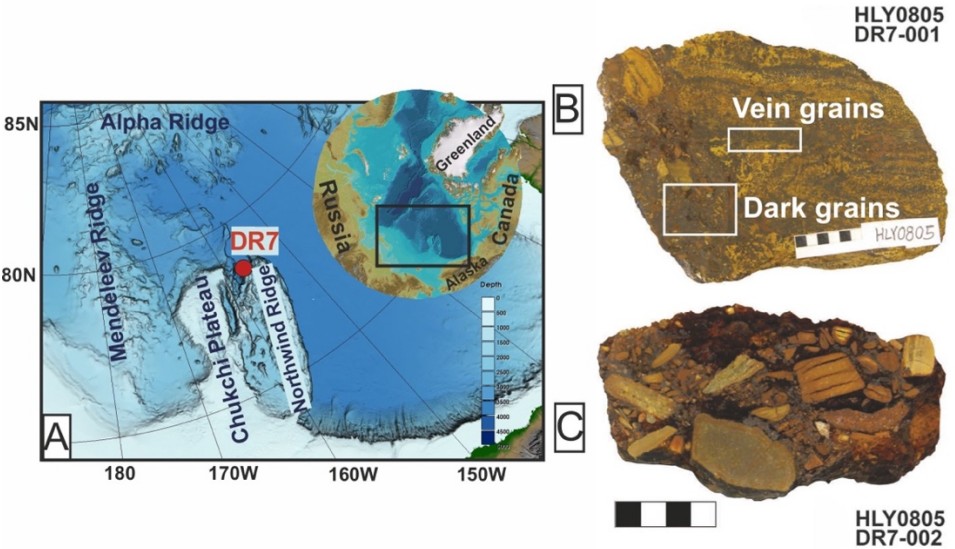

**Figure 1. (A) Regional setting of the Amerasia Basin (inset) and location map of the DR7 dredge haul; (B) cut section images of the main sample types; all subsamples for age dating are from DR7-001**

The dominant mineral in the mineralized samples based on X-ray diffractions is goethite and possibly lesser amounts of feroxyhyte ($\delta$-FeOOH) and ferrihydrite [$Fe^{3+}_{4-5}(OH,O)_{12}$] (Table 1; Fig. 2). The darker colored goethite has better crystallinity than the paler ones.

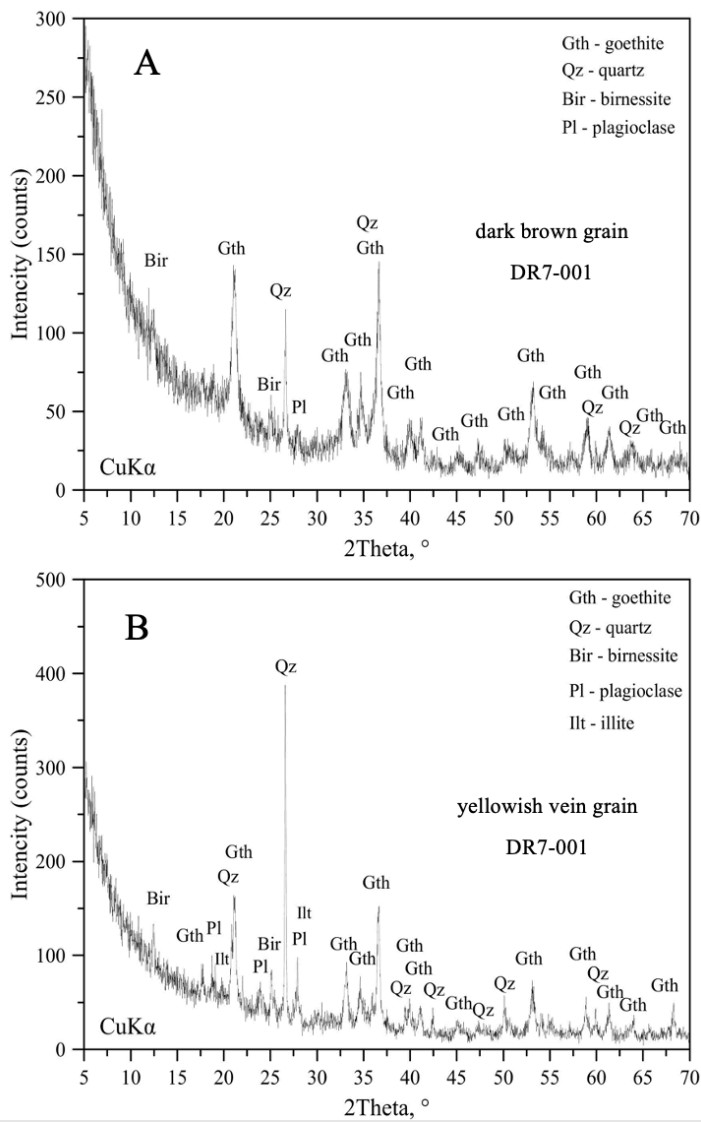

**Figure 2. X-ray diffraction mineralogy of the dark brown clast (A), and yellow-brown vein material (B) from the DR7-001 sample.**

Based on SEM-EDS studies (Fig. 3) Fe-oxides crystallite sizes of cement and replacements vary from submicrometer to a few micrometers, rarely up to 120 μm. Birnessite and 10Å manganates (todorokite, buserite, or asbolane) and δ-MnO$_2$ (vernadite) occur as well. Relict host-rock minerals include quartz, feldspar, mica, and clay minerals. Clinochlore (chlorite) is ubiquitous in the DR7 samples. Among the U-bearing minerals, single grains of zircon and monazite were observed (Fig. 3).

**Table 1. XRD Mineralogy of Crystalline Phases of DR7 Sample from the Amerasia Arctic Ocean.**

| Sample ID | Description | Major | Moderate | Minor |
|---|---|---|---|---|
| DR7-001-L1A | Cement from breccia | Goethite | Quartz, Birnessite | TAM |
| DR7-001-L1B | Glassy Fe-rich clast | Goethite | -- | -- |
| DR7-001-L2B | Fe-rich dark-brown lamina | Goethite, Birnessite | Clinochlore, Quartz, Plagioclase | δ-MnO$_2$, TAM |
| DR7-001-L2D | Reddish vein | Goethite, Quartz | Clinochlore, Plagioclase, Mica | Birnessite, TAM(?) |
| DR7-001-L2E | Fe-Mn lamina | Birnessite, Goethite | TAM(?), Quartz | Plagioclase |

Major >25%, Moderate 5-25%, Minor <5%. TAM is 10Å Manganates = todorokite, buserite, or asbolane. Goethite may also include feroxyhyte or ferrihydrite.

Comment: X-ray diffraction mineralogy was completed using a Malvern Panalytical X'Pert Powder X-ray diffractometer (XRD) with CuKα radiation and graphite monochromator run from 4° to 70° 2θ with a step size of 0.02° 2θ at 40 kV and 45 mA at USGS, PCMSC lab in 2017. Digital scans were analyzed using Philips X'Pert High Score Plus software to analyze X-ray reflections and identify possible mineral phases.

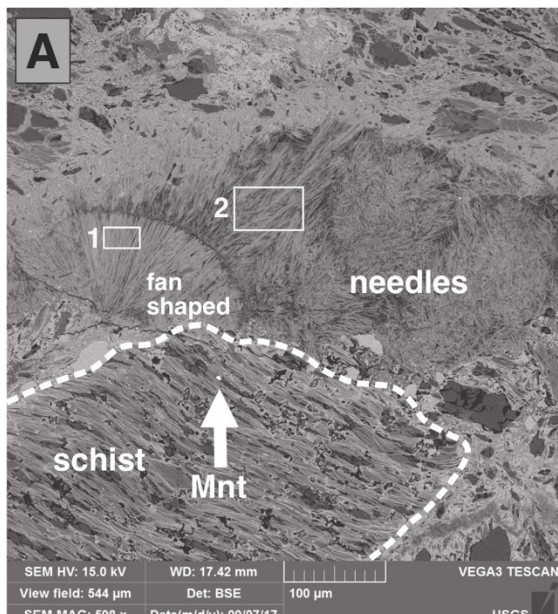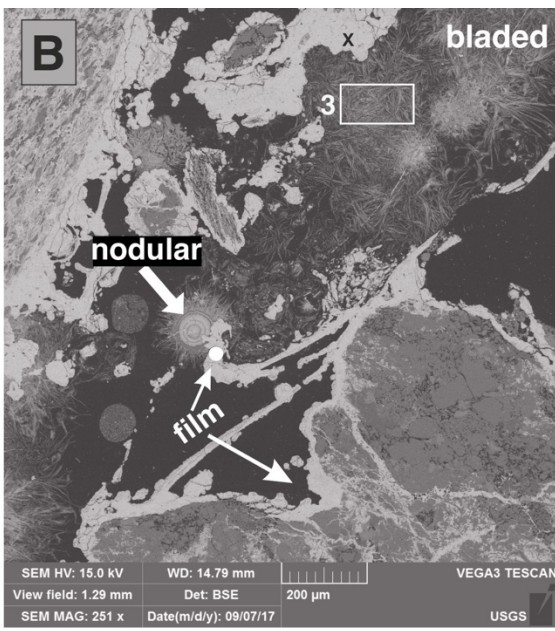

**Figure 3. Backscatter SEM photomicrograph images of sample DR7-001 amd DR7-002 from polished thin sections; (A) fan-shaped (26% Fe, 17% Mn for box 1), needle (28% Fe, 15% Mn for box 2), and massive cements of Fe and Mn oxides in the breccia part of sample DR-001; note schist grain in the lower left quadrant with a bright monazite grain (Mnt); (B) cement of breccia: bladed (box 3: 29% Fe, 35% Mn), nodular (white arrow: 33% Fe, 25% Mn), and film type (white dot: 68% Fe; black x: 35% Fe, 29% Mn) Fe- and Mn-oxide cements; bladed cement consists of discrete Mn-oxide and Fe-oxyhydroxide blades, and Fe and Mn contents vary for each laminae in the micronodule.**

**Polished thin sections were carbon coated and used for SEM-EDX analyses of samples DR7-001 and DR7-002 using a Tescan Vega3 scanning electron microscope (SEM) at operating conditions of 30 kV and 15 nA for imaging; the Energy Dispersive Spectrometry (EDS) chemical characterization and element mapping was done using a JEOL 8900 operating at 15 kV and 40 nA for quantitative analyses of oxides; counting times were 30 s peak and 15 s background at USGS lab in Menlo Park in 2017.**

## 3. Methodology and sampling strategy

For (U-Th)/He dating fragments of goethite mineralization were manually extracted from the DR7-001 sample: dark-brown clasts of breccia and yellow-brown vein material (Fig 1B). According to the SEM and XRD data samples mainly consist of pure crystalline goethite (Fig. 2,3) with possible admixture of birnessite and quartz. Therefore, the samples represent a standard material which is used for (U-Th)/He dating. But samples themselves are not typical for (U-Th)/He studies. Most of the goethite grains that are used for He geochronology are from terrestrial supergene environment (Monteiro et al., 2024). The low-

temperature steady and deep underwater environment (~ 0°C) prevent thermal loss of He from the analysed samples during their geological history.

### 3.1. Sample preparation

In order to exclude possible U-loss during the sample preparation when goethite grains are sonicated in a distilled water the leaching experiments were conducted. Millimetre-size fragments of goethite were manually extracted from the DR7-001 sample which represented dark-brown clast of the breccia and a yellow-brown vein material (Fig. 1B,C). At the first stage the massive single fragments in the closed Teflon vials with 3 ml of distilled deionized water (Barnsted) were sonicated for 15 min at room temperature (the temperature was not stabilised by extra cooling). The solution was removed by the mechanical pipette for subsequent chemical analyze. At the second stage the remained grains were dried at room temperature for 24 hours and crushed in the Teflon vial by the molybdenum stick ( < 300 μm) to increase their specific surface area. The crushed grains were sonicated in distilled deionized water (Barnsted; 3 ml) for extra 15 min at room temperature without extra cooling. After the solutions were left for 24 hours for the sinking of the small floating particles. The uppermost 1 ml of the solution was carefully moved to a new beaker and nitric acid was added up to 5% $HNO_3$ solutions (50–150 μl). Uranium and Th contents were measured by ELEMENT 2 ICP mass-spectrometer at the Institute of Precambrian Geology and Geochronology RAS. The full procedural blanks were obtained by the parallel procedures with an empty beaker. The total U and Th content of the sample was determined in the same way after its complete dissolution in the mixture of aqua regia (200 μl) with HF (250 μl) and $HClO_4$ (10 μl) for 15 h at 110° C in a closed Teflon vial in thermostat. Due to described analytical procedure the obtained U and Th contents in the leaching solutions are semi-quantitative.

### 3.2. (U-Th)/He dating

Eight millimeter-size fragments of goethite mineralization were manually extracted for (U-Th)/He dating from three different parts of the DR7-001 sample: two dark-brown clasts of the breccia and a yellow-brown vein from the completely altered rock (Fig. 1B,C; Table 2). Subsamples from the yellow-brown vein material and from dark-brown gains were treated as separate samples (1-8). Samples were derived from the inner part of the original sample, had no visible under the binocular microscope inclusions of other minerals and were not washed.

### 3.2.1. Measurement of radiogenic [4]He contents

For each measurement, ~1–3 mg fragments of goethite grains were placed in a quartz ampoule (~1 cm long) and sealed under a $10^{-3}$ torr vacuum (Fig. 4). The sealing was done by the distilled water-based torch LIGA (Vasileostrovsky Electrochemical Plant). The torch has a narrow flame that prevent heating of the sample during the sealing. The Durango apatite (n=3) sealed in a quartz ampoules by the same technique did not show any sign of He loss (Fig. S1).The sealed ampoule, via a special gateway, was placed in a high-temperature high-vacuum furnace of the magnetic sector MSU-G-01-M mass-spectrometer

equipped with two SAES getter pumps (Spectron Analyt, IPGG RAS; Shukolyukov et al. 2012a,b). During heating, He easily diffuses through the thin quartz walls while U and other products of the sample decomposition remain in the ampoule. A Secondary Electron Multiplier (SEM) was used to determine the $^4He^+$ beam intensity (cps). Calibration of the mass spectrometer was done using Knyaghinya meteorite (Schultz and Franke, 2004) and RS-Pt reference material (Yakubovich et al. 2023).

Goethite samples were step-heated at temperatures of 350° C for 30 min, 550° C for 10 min, 900° C for 10 min, 1100° C for 15 min, and 1150° C until He stop to release (5 min in average). Samples 1 and 2 (ID 966, 969, Table 2) were step-heated under slightly different conditions, starting with a temperature of 240° C. This step-heating approach allows for monitoring the He release pattern from the goethite grains as well as the excess hydrogen (ion $HD^+$) in the chamber of the mass-spectrometer. The diffusion of He through the thin quartz walls of the ampoule is fast (Shuster and Farley, 2005; Yakubovich et al., 2014), but it does not allow to obtain the accurate diffusion kinetics of He from the goethite grains.

Following the extraction of He, the ampoule was removed from the mass spectrometer for subsequent separation of U and Th. The total procedural blank, determined by heating the empty quartz ampoules to 1100°C, corresponds to $4.4 \pm 1.6 \times 10^{-10}$ cm$^3$ He at STP.

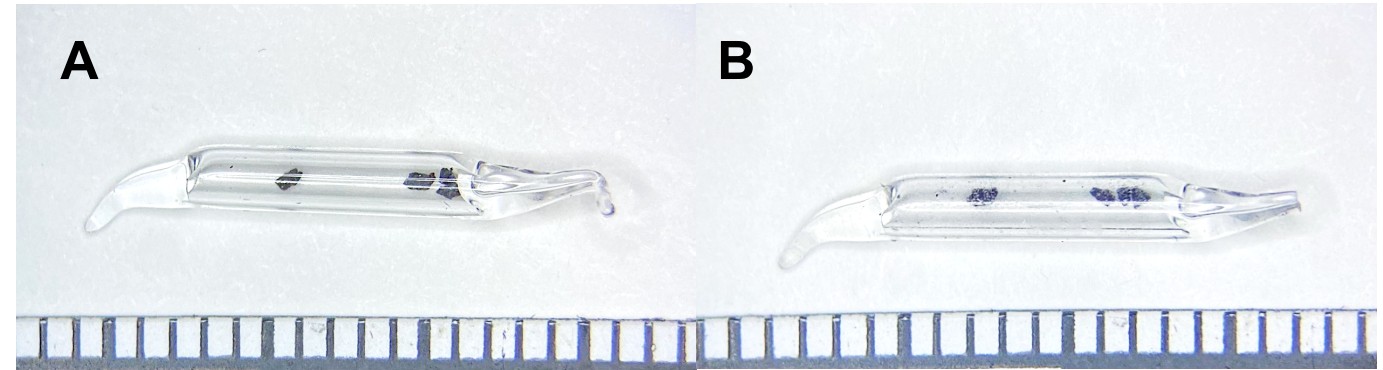

Figure 4. Fragments of goethite in a sealed quartz ampoule (A) before heating; (B) after heating. Scale bar 1 mm.

### 3.2.2. Measurement of U and Th Contents

The quartz ampoule with degassed samples was spiked with a $^{230}$Th-$^{235}$U tracer and dissolved in a mixture of aqua regia (0.4 mL), concentrated hydrofluoric acid (0.5 mL), and perchloric acid (0.05 mL) in closed Teflon vials for 2 hours at 200° C on a hot plate followed by 15 h at 110° C in a thermostat. The solution was dried on a hot plate at 200° C. During this step, perchloric acid prevented the formation of low-soluble fluorine complexes, while most of Si evaporated in a form of SiF$_4$. The remaining precipitate was dissolved in 1.5 mL of 5% nitric acid and heated up to 80° C in an ultrasonic bath for 15 min prior the

measurement of U and Th contents. $^{235}U/^{238}U$ and $^{230}Th/^{232}Th$ isotope ratios were measured on an ELEMENT XR ICP mass-spectrometer at the Vernadsky Institute of Geochemistry and Analytical Chemistry RAS. The total chemical procedure blank, determined by dissolution of the empty quartz ampoules (n=4) using the same settings, corresponds to 1.30 ±1.26 and 5.8±4.4 $10^{10}$ atoms of $^{238}U$ and $^{232}Th$ respectively.

The (U-Th)/He ages were calculated using IsoplotR software (Vermeesch, 2018). The combined analytical uncertainty was estimated based on the U, Th, and He measurement uncertainties and the uncertainty based on the blank determinations. The alpha-recoil corrections were not applied, because all analyzed samples are the fragments of large grains.

**Table 2. Results of (U-Th)/He Dating of Goethite Subsamples of DR7-001**

| No. | ID | Type | Mass [mg] | U [ppm] | U $[10^{10}$ at] | σ [%] | Th [ppm] | Th $[10^{10}$ at] | σ [%] | Th/U | $^4He^d$ [cm³ STP g⁻¹] | $^4He$ $[10^{10}$ at] | σ [%] | age [Ma] | 2σ |
|---|---|---|---|---|---|---|---|---|---|---|---|---|---|---|---|
| 1 | 966 | Dark grains | 0.954 | 2.51 | 603 | 1.8 | 0.81 | 194 | 3.6 | 0.3 | 2.9 x 10⁻⁶ | 7.54 | 2.3 | 9.1 | 0.5 |
| 2[a] | 969 | | 7.197 | 1.86 | 3376 | 1.1 | 0.69 | 1258 | 2.1 | 0.4 | 3.4 x 10⁻⁶ | 65.17 | 2.8 | 13.8 | 0.8 |
| 3 | 1015 | | 1.946 | 2.62 | 1287 | 1.3 | 0.66 | 323 | 1.8 | 0.3 | 2.6 x 10⁻⁶ | 13.73 | 3.7 | 7.9 | 0.6 |
| 4 | 1022 | | 1.908 | 2.78 | 1338 | 1.8 | 0.81 | 387 | 2.0 | 0.3 | 3.1 x 10⁻⁶ | 15.97 | 3.2 | 8.7 | 0.6 |
| 5[a] | 1031 | | 2.973 | 2.43 | 1823 | 2.8 | 1.99 | 1491 | 2.4 | 0.8 | 4.3 x 10⁻⁶ | 33.99 | 2.8 | 12.2 | 0.9 |
| 6 | 1032 | | 3.115 | 2.25 | 1769 | 5.2 | 1.76 | 1379 | 5.0 | 0.8 | 2.6 x 10⁻⁶ | 21.85 | 2.7 | 8.2 | 0.8 |
| | | | | | | | | | | | | Dark brown grains weighted mean[c] | | | 8.6 ±0.3 | |
| 7 | 1033 | Vein grains | 1.782 | 1.36 | 613 | 2.2 | 3.30 | 1481 | 1.6 | 2.4 | 1.1 x 10⁻⁶ | 5.40 | 8.7 | 4.4 | 0.8 |
| 8 | 1036 | | 2.708 | 1.80 | 1232 | 2.5 | 4.90 | 3347 | 1.9 | 2.7 | 1.8 x 10⁻⁶ | 12.70 | 3.9 | 4.9 | 0.4 |
| | | | | | | | | | | | | Yellowish brown vein weighted mean[c] | | | 4.8 ±0.4 | |
| | Empty Quartz ampoule[b] | | 28–56 | -- | 1.3 | 97 | -- | 6 | 74 | -- | -- | 1.1 | 37 | -- | -- |

The reported uncertainties of U, Th, and He measurements are the combined uncertainties calculated by summation in quadrature of measurement and blank uncertainties using a coverage factor of 1 which gives a level of confidence of approximately 65%.

[a]No. 2 and 5 ages not used in the calculation of mean age; see text in Results section for explanation.

[b]Contents of U, Th, and He in the quartz ampoule represent full analytical blank, which includes chemistry and steps of sample preparation (sealing, heating).

[c]The reported uncertainties of an age value is an expanded analytical uncertainty which include analytical uncertainty of U, Th, He measurements and factors addressed at the section 4.2. Error value corresponds to 95% level of confidence (2σ).

[d]The calibration of the mass-spectrometer was done using the mineral reference materials in a range of He content from 2 to 200 [$10^{10}$ at].

**Table 3. Results of the leaching experiments of Goethite Subsamples of DR7-001**

| Sample | Stage | Weight, mg | U, ng | Th, ng | Th/U | Fraction of U-loss | Fraction of Th-loss |
|--------|-------|-----------|-------|--------|------|--------------------|---------------------|
| dark grain | first | 5.628 | 0.03 | 0.01 | 0.26 | 0.3 | 0.3 |
| | second | | 0.38 | 0.02 | 0.06 | 3.5 | 0.6 |
| | residual | | 10.7 | 3.4 | 0.32 | - | - |
| dark grain-2* | second | 2.462 | 0.03 | 0.02 | 0.6 | 0.5 | 0.23 |
| | residual | | 5.4 | 7.3 | 1.3 | | |
| vein grain | first | 6.212 | 0.01 | 0.02 | 1.4 | 0.12 | 0.10 |
| | second | | 0.6 | 0.40 | 0.6 | 7.8 | 1.6 |
| | total | | 8.4 | 24.7 | 2.5 | - | - |
| vein grain-2* | second | 1.890 | 0.11 | 0.19 | 1.7 | 3.0 | 1.7 |
| | residual | | 3.6 | 10.9 | 3.0 | | |
| blank | first | - | 0.004 | 0.002 | 0.5 | - | - |
| | second | - | 0.01 | 0.005 | 0.7 | - | - |
| | residual | - | 0.01 | 0.02 | 1.8 | - | - |

Comment: * grains were crushed and sonicated without previous step (first stage).

## 4. Results

### 4.1. Leaching experiments

Chemical analyses of the distilled water leachates revealed the partial loss of U and Th from the subsamples (Table 3). The leaching of U and Th from the crushed subsample is more intensive than from a massive grain and reach up to 8% for U and less than 2% for Th. Because the samples are not water soluble but the leachates contain also some amount of Mn, Fe and Co, some tiny floating particles of the original sample might be in the solution. The ICP-MS measurements were calibrated only for U and Th, thus determining the concentration of Mn and Fe in the solutions was not possible. However, the notable shift of the Th/U ratio in the solution relative to the Th/U ratio of the residual goethite (from 0.06 to 3; Table 3) indicates that some part of U was leached from the samples. These findings are in an agreement with previous results of Fe- and Mn-oxides leaching experiments by a weak acids with acetate buffer (Konstantinova et al., 2018; Koschinsky and Hein, 2003) which implies U and Th adsorbed behavior.

### 4.2. (U-Th)/He dating results

The (U-Th)/He ages for eight fragments of goethite from sample DR-7-001 included fragments of two sets of dark grains from two separate breccia goethite clasts (grains 1-4 and 5-6) and one set of yellow-brown vein samples (grains 7-8) (Table 2; Fig. 5). The signals of He, U, and Th of all samples were markedly higher than the background level (empty quartz ampoule). The concentrations of U in the dark goethite grains range from 2.2–2.8 ppm, with Th/U ratios of 0.3–0.8. The concentration of U

in the two vein subsamples is lower (1.36 and 1.8 ppm) and Th prevails over U (Th/U 2.4–2.7). Concentrations of $^4$He range from 2.6 to 4.3 x 10$^{-6}$ cm$^3$ STP g$^{-1}$ for the dark-brown grains and from 1.1 to 1.8 x 10$^{-6}$ cm$^3$ STP g$^{-1}$ for vein samples. Among
the six dark goethite grains analyzed, one had an atypically low U concentration (1.86 ppm; Table 2). Sample 5 (ID 1031) had an unusual high-temperature He release pattern (>1100 °, Fig. 7), which likely indicates the presence of He-retentive mineral inclusions. These anomalous samples (ID 969; 1031) revealed (U-Th)/He age in a range of 12.2–13.8 Ma (Table 2). The coincidence of their (U-Th)/He ages might indicate that we were wrong when decided to exclude these grains from the consideration. However, in a lack of confidence we are not going to interpret this age value.

The (U-Th)/He age of the remaining dark grains is consistent within the uncertainty of the measurements with a weighted mean value of 8.6±0.3 Ma (2σ). The two yellow-brown vein samples had significantly younger reproducible ages, with a mean of 4.8±0.4 Ma (2σ).

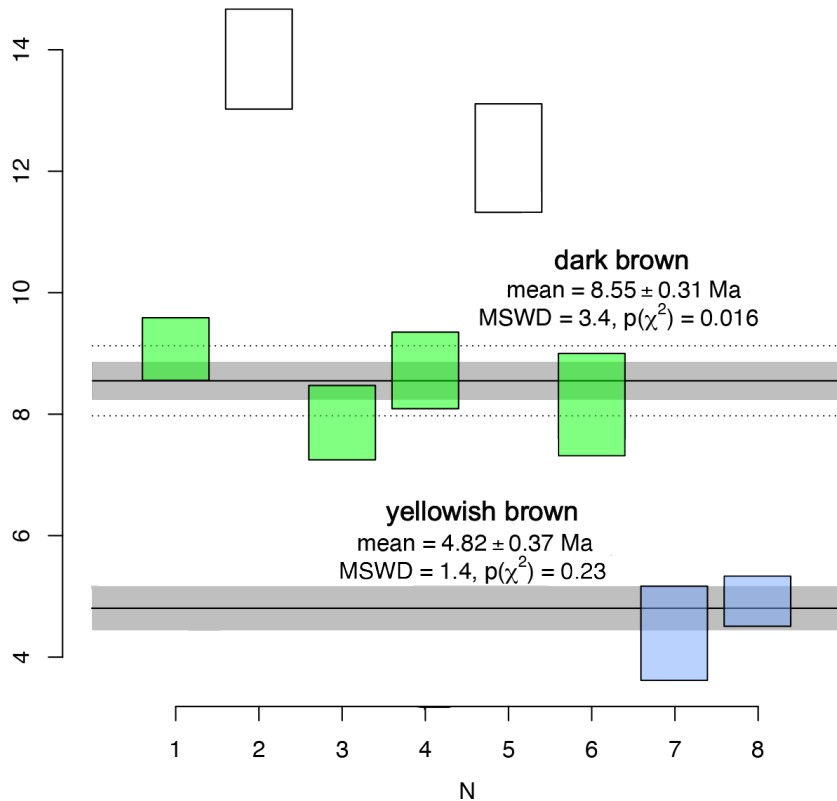

**Figure 5. Results of (U-Th)/He dating of goethite from DR7-001 subsamples. Error bars 2σ. Weighted mean plot constructed using IsoplotR software (Vermeesch, 2018).**

## 5. Discussion

### 5.1. Methodological implications

#### 5.1.1. Sample preparation

Due to the leaching experiments around 8% of U and 1.7% of Th can be remobilised from the sample by the fresh deionized distilled water, which is known to became chemically active after the contact with atmosphere (pH 5–6; Gurr, 1962). Goethite is not a water-soluble mineral therefore U release likely indicates its position beyond the crystal lattice or in some unstable phases. Th/U ratios of the grains (0.3 and 1.3 for dark grains; 2.5 and 3.0 for a vein material) are remarkably higher than those of leachates (0.06 and 0.6; 0.6 and 1.7, respectively; Table 3), which implies that U is easier to mobilize. This is in favor of

the adsorbed form of some of the U, rather than the presence of unstable phases with different Th/U ratios. The higher percent of U-loss from the crushed samples is also in agreement with this suggestion (Table 3).

The possible adsorbed behavior of U in goethite from the weathering environment was discussed by Shuster et al (2005) and Vasconcelos et al. (2013). The leaching experiments are also in agreement with the results of radiochemical experiments that revealed that during the crystallization of hematite and goethite from ferryhydrite $(Fe^{3+})_2O_3 \cdot 0.5H_2O$), which is the least stable

iron (oxyhydr)oxide, only part of uranium becomes leaching-resistant (Payne et al., 1994).

The proportion of U in adsorbed form relative to the U, which is incorporated into crystal lattice can differ from sample to sample. This is indirectly confirmed by the discussion in Vasconcelos et al. (2013), which suggest that various patterns of U-loss during the He release from the goethite samples possibly indicated different U position of the analyzed samples. Adsorbed behavior of some of the U does not affect strongly on applicability of the (U-Th)/He method due to the long alpha-stopping

distances (Shuster et al., 2005). However, sonication of the samples in distilled water prior (U-Th)/He dating might result in U-loss and subsequent erroneous/over-dispersed ages of multi-mineral grains. Large grains (crystals > 50-70 mkm) are unlikely to lose the significant amount of U as their surface to volume ratio is low.

#### 5.1.2. Justification of the technique

(U-Th)/He ages of goethite subsamples are reproducible. Measured concentration of U in the samples which were degassed in the quartz ampoules  (1.4–2.8 ppm; Table 2)  are in a range of their concentrations in the unheated grains (1.5–2.5 ppm; n=5; ICP MS). This indicates that the proposed analytical approach is well suited for (U-Th)/He dating of goethite, and likely other Fe-(oxyhydro)oxides. Encapsulating the individual goethite grains into the quartz ampoule exclude any U-loss during the sample degassing which is one of the major analytical concerns (Vasconcelos et al., 2013; Hofmann et al., 2020; Wu et al.,

2019). The approach allows overheating of the sample with plenty of reserve. Based on our experience on He release from isoferroplatinum (Pt₃Fe) quartz  ampoule are robust to temperatures up to 1450°C (Shukolyukov et al., 2012b). One of the

main disadvantages of the proposed technique is the relatively high blank of the quartz ampoule, which complicates analyse of very small and/or grains that are too young. The technique is quite sufficient for (U-Th)/He dating of mg-weighted samples of Neogene age as tested here, and require remarkably lower amount of the material than double- or multi aliquot approaches (Pidgeon et al., 2004; Wu et al., 2019). The technique also does not require the modernisation of the He extraction line which is needed for the degassing in the presence of $O_2$ (Hofmann et al., 2020).

### 5.1.3. Future developments

In order to determine the analytical limitations of the proposed methodological approach additional tests and improvements should be done in the future. The technique is based on several key assumptions. The first assumption (a) is that no He loss occurs from the sample when sealing a quartz ampoule with a torch. Several experiments support this assumption, conducted on Durango apatite (Fig S1) and more He-retentive minerals such as isoferroplatinum (Yakubovich, 2013) and pyrite (Yakubovich et al., 2020b). However, it would be necessary to date by this technique Fe-(oxyhydro)xides with independent age constraints in order to confirm the suitability of the procedure. Measurement of a comprehensive set of Durango apatite grains by the same approach is another possible way to confirm the absence of He lost by the grains during the sealing procedure.

The second assumption (b) is that heating at 1150°C is sufficient to release all He from the goethite grains. This is based on the observed He release pattern, but further verification is needed. Future experiments should involve heating goethite to higher temperatures to confirm the complete release of He.

The third assumption (c) is that the quartz ampoule prevents U loss. This is based on the relatively slow diffusion of uranium (U) in quartz. However, to validate this, it is necessary to compare the results with U and Th measurements from unheated aliquots to demonstrate complete recovery of volatilized U. Since goethite is opaque and mineral inclusions could affect (U-Th)/He ages, pre-screening the grains using micro-CT would be important for these tests and would enhance the proposed methodological approach.

This study also identified that U can leach from goethite during sonication in distilled water. The hypothesis of U mobilization from goethite is primarily based on the shift in the Th/U ratio in the solution compared to the residual goethite (Table 3). However, this observation is indirect, as other factors, such as intrinsic variability in the Th/U ratio of the goethite, may contribute to this shift. To confirm U mobilization, a series of experiments comparing the (U-Th)/He ages of washed and unwashed goethite grains is necessary. It remains unclear whether this U behavior is specific to underwater hydrothermal multi-mineral goethite or whether a similar pattern could be observed in terrestrial supergene goethite.

### 5.2 Geological implications

The results of the (U-Th)/He age dating of goethite grains from the slope of Chukchi Borderland produce a Neogene age formation. There are several factors that might potentially affect the mineral age results, such as He loss, radiation damage, recrystallisation, and fluid and mineral inclusions, which we discuss below.

280 ### 5.2.1. Helium thermal retentivity

Goethite is predominantly He retentive under surface conditions (Cooperdock and Ault, 2020). The mineral is able to retain around 80–95% of its radiogenic $^4$He for millions of years (Shuster et al. 2005; Deng et al. 2017; Hofmann et al., 2017). The water temperature at 3400 m water depth within Chukchi Borderland slope is about -0.3 °C (Zhang et al., 2021), therefore any thermal loss of He seems unlikely, though it could be induced by local hydrothermal events.

285 Heating the sample in a quartz ampoule does not allow the measurement of the He diffusion parameters, nevertheless it does reflect the He retentivity of the sample. Despite different He release patterns (Fig. 6) the (U-Th)/He age of the same group is quite uniform, which likely indicates insignificant thermal loss of $^4$He (Fig. 6). Remarkable that He release pattern of the sample 3 (ID 1015) significantly differs from the patterns of the other grains, despite its (U-Th)/He age is consistent with other measurements.

290

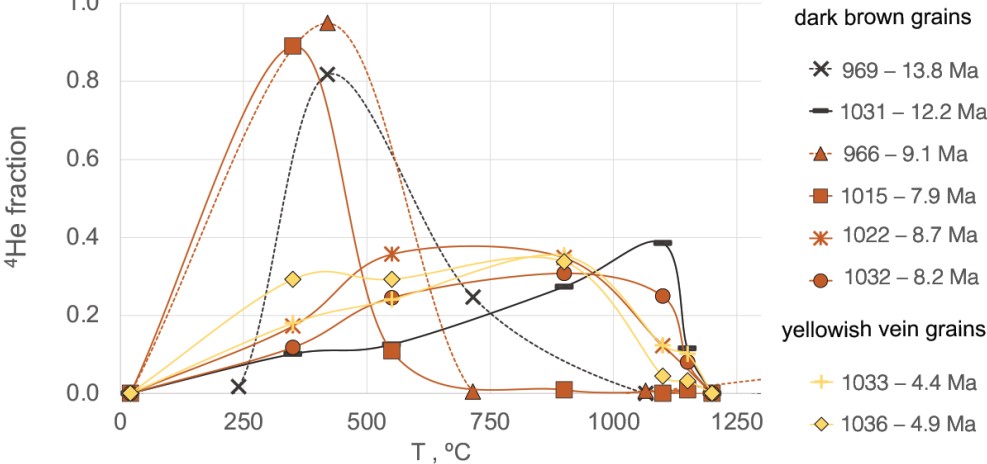

**Fig. 6. Helium release pattern from the mainly (>95%) goethite grains sealed in Quartz ampoule. All measurements (with the exception of the samples 966 and 969; dotted line) were carried out under the same time-temperature conditions. Values are a sample ID in the Table 2.**

### 5.2.2. Radiation damage

The He diffusivity of a mineral can be strongly affected by the amount of accumulated radiation damage (Flowers et al., 2023). The standard technique that is used to investigate the role of radiation damage and elemental substitution, is not applicable to goethite due to its dehydration during vacuum step-heating (Farley et al., 2023). Numerical simulations, which combined the Density Functional Theory (DFT) and Kinetic Monte Carlo (KMC) simulations predicts that He loss from goethite is strongly controlled by radiation damage and some other impurities (e.g., Al) (Bassal et al., 2022).

The samples has close values of eU content and in the limited range of its variation there is no correlation with (U-Th)/He age values (Fig. 7). The uniform (U-Th)/He ages of the petrological groups (clasts and vein) indicate limited impact of the radiation damage on the dispersion of He ages.

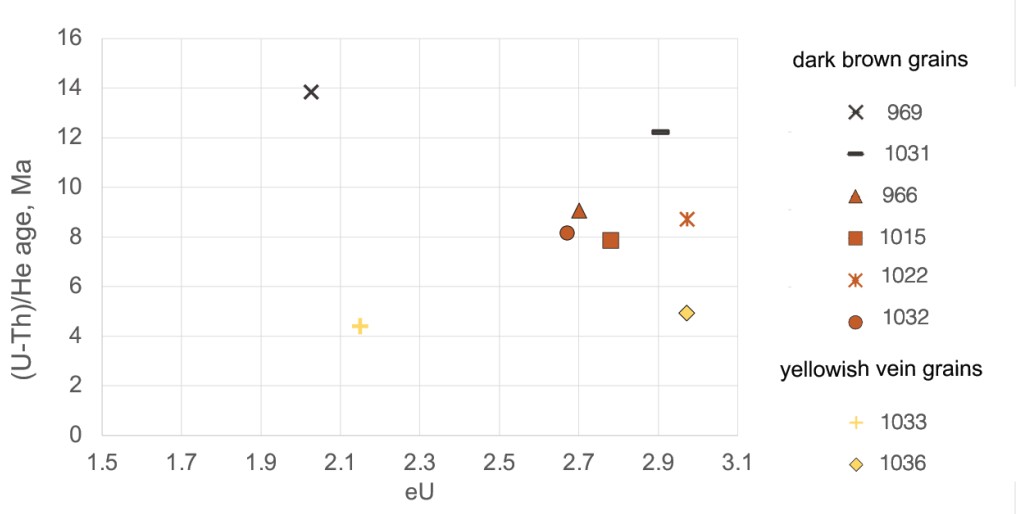

**Figure 7. Goethite (U-Th)/He ages versus eU concentration of the samples. The effective modern uranium concentration (eU) was calculated based on the formulas given by (Flowers et al., 2023). Index is a sample ID in the Table 2.**

### 5.2.3. Mineral and fluid inclusions and impurities

The studied samples contain rare U- and Th-rich mineral inclusions, such as zircon and monazite, with sizes ranging between < 1 to 40 µm (e.g. Fig. 3). If there was incomplete dissolution within the chemical procedure, the U-loss would result in an erroneously old and unreproducible ages, which might be the case of sample 5 (ID 1031).

Helium concentration of minerals fluid inclusions that formed during magmatic and hydrothermal processes typically does not exceed $10^{-8}$ cm$^3$ STP g$^{-1}$ (Stuart et al., 1994; Graupner et al., 2006), that is less than 1% of the total He of the studied samples and insignificant for our (U-Th)/He dating procedure.

Incorporation of Sm can be an additional source of $^4$He in goethite. Sm contents of the DR7-001 samples vary from 5.5 to 6 ppm (ICP-MS data; lithium-metaborate fused disks; n=3), which implies that Sm would produce less than 0.25% of the He sample budget.

XRD data indicates that goethite from the yellow-brown vein material has lower crystallinity and higher abundance of mineral inclusions such as quartz, plagioclase and illite (Fig. 2; Table 1). These factors potentially might decrease the (U-Th)/He age of the samples. Thus, we cannot exclude that the younger age of the goethite from the yellow-brown vein is related to some He-loss.

### 5.2.4. Recrystallization

Goethite is the most thermodynamically stable Fe-(oxy)hydroxide in the near-surface environment. However, it can undergo dissolution–recrystallization processes during interaction with acidic solutions that reset its (U-Th)/He age (Monteiro et al., 2014). These processes enrich samples with low-soluble components that increase the Th/U ratios. This might be initiated by the presence of $Fe^{2+}$ ions in the aquatic systems (Handler et al., 2014). Given that the vein has higher Th/U ratios (2.5–2.7 vs 0.3–0.8 of the dark grains) and younger (U-Th)/He age (4.8±0.9 Ma vs 8.6±1.2 Ma; Table 2), its newly formation due to the recrystallization of goethite cannot be ruled out.

### 5.2.5. Interpretation of (U-Th)/He ages

In additional to assessment of the all factors that might impact the (U-Th)/He ages, we include 10% (2σ) uncertainty to the primary analytical uncertainty of the measurements based on the suggestion of (Monteiro et al., 2014). Thus, the dense dark-brown goethite have the age of 8.6±1.2 Ma (2σ), and the vein material is younger, 4.8±0.9 Ma (2σ). These values do not overlap within the extended uncertainty.

The (U-Th)/He ages reflect the time of mineral formation, recrystallization, or cooling below the closure temperature. Closure temperature of goethite varies over a wide range, from ~20 to 150° C, depending on the diffusion domain sizes and distribution of the defects in the crystal lattice (Bassal et al., 2022). Thus, the uniform (U-Th)/He ages of the dark-brown grains accompanied by remarkably different He release patterns (Fig. 6) might be explained by cooling, only with the assumption of fast (1–2 Ma) host rocks uplift from ~ 2–4 km depth that took place ~9 Ma ago. However, that assumption is inconsistent with the tectonic evolution of the Arctic region (e.g., Chian et al., 2016; Craddock and Houseknecht, 2016). Therefore, (U-Th)/He ages of the dark grains of pure crystalline goethite reflect a Neogene mineralization event in the Chukchi Borderland, Arctic Ocean. More data is required in order to check the possible presence of fragments of older Fe- and Mn- mineralised rocks (12–14 Ma; Table 2), as well as to confirm the young (~ 4.8 Ma) mineralization event.

## 6. Conclusion

Reproducible (U-Th)/He ages is achieved using our proposed analytical approach, which involves sealing the sample in quartz ampoule for He release is well suited for (U-Th)/He dating of Fe-(oxy)hydroxides; this techniques allows for the determination of U, Th, and He on the same subsample aliquot. Our data also indicate a fraction of U can be leached from multi-grain goethite samples during sonication in the distilled water, implying that this step of goethite sample preparation for (U-Th)/He dating should be taken with caution.

(U-Th)/He ages of goethite from the slope of the Chukchi Borderland formed during a Neogene mineralization event (8.6±1.2 Ma). The younger age of the yellow-brown vein material (4.8±0.9 Ma) can be explained by an episode of later-stage mineralization, recrystallization or by its lower crystallinity. Further investigations and a larger sample set are recommended for a comprehensive understanding of the geological evolution of the region.

## Competing interests

The contact author has declared that none of the authors has any competing interest.

## Acknowledgements

We thank James R. Hein and Kira Mizell from U.S. Geological Survey for the provision of samples used in this study and for reviews. We are grateful to Irina Volkova (Department of crystallography; Saint Petersburg State University) for the assistance. Xiao-Dong Deng, Florian Hofmann, Hevelyn S. Monteiro, Marissa Tremblay, Cecile Gautheron are gratefully acknowledged.

This research was supported by RSF 22-77-10088. Chemical analyses done by M.O. Anosova were funded by the State Assignment of the Vernadsky Institute of Geochemistry and Analytical Chemistry, Russian Academy of Sciences.

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
