# Peer review of "An analytical approach for (U-Th)/He dating of goethite by sample encapsulation in quartz ampoules under vacuum, with an example from the Amerasian Basin, Arctic Ocean"

_EGUsphere, 2024_

## Author Comment (AC1)

Sequential leaching is a technique to group elements with their host mineral phase or with a group of minerals. It consists of four steps. (1) Exchangeable cations and calcium carbonates: 1 g of powdered sample was stirred with 1 M acetic acid/Na acetate buffer (pH 5); (2) Easily reducible fraction: The residue of leach 1 was stirred with 0.1 Mhydroxylamine hydrochloride solution (pH 2); (3) Moderately reducible fraction: 0.2 M oxalic acid/ammonia oxalate buffer (pH 3.5) was added to the solid residue of step 2; (4) Residual fraction: The residue from step 3 was digested in Teflon bombs in a microwave, using a mixture of 48% HF and 65% HNO3 (proportion 2/1) and dried; then 25% HCl was added and dried again. All steps were carried out twice. Hydrogen peroxide was used to dissolve organic matter.

Sequential leaching experiments were carried out in the laboratory at the USGS, Pacific Coastal and Marine Science Center. Al, As, Ba, Ca, Cd, Co, Cr, Cs, Cu, Fe, K, Li, Mg, Mn, Ni, Pb, Ti, Th U, V, Zn, and Y were measured in the four phases using an ICP-MS PerkinElmer NexION 300Q (USGS, Menlo Park, operators N. Konstantinova). For each element, the sum of the amounts released by leaching steps 1- 4 was compared with the bulk analysis data to check recovery, which was between 80% and 120%.

---

## Author Comment (AC2)

[Figure]

**Fig. 3.** Kinetics of [4]He release from a native gold sample from the Karalon deposit, Republic of Buryatia. (a) Appearance of the sample after heating up to 850°C (no changes); (b) sample after heating up to 1000°C (incipient melting); and (c) sample after heating up to 1100°C (complete melting and formation of a sphere).

Image from Yakubovich et al., 2014 which shows He release pattern from the native gold. You can see that He content is constantly rising with the exception of the last heating step (1100 C).

[Figure]

Sup 2 Fig 2

Helium spectra from sample ID 1036

That is how the measurement results look like. The second peak is 4He. The first one is mass -3 which are mainly hydrogen molecules. The first peak and the background itself reflects the cleanness of the sample (water/trapped gases content).

You can see that He content of the sample was constantly growing till the scan number 60. And than He content was stable for other 10 scans. Heating was off at scan number 72. Last 10 was used to calculate the average He content of the sample.

---

## Author Response (AR1)

Dear Marissa,

We are pleased to submit the revised version of the original manuscript "An analytical approach on (U-Th)/He dating by sample encapsulation in quartz ampoules under vacuum: application to goethite, Amerasian Basin, Arctic Ocean".

We have taken into account all your suggestions, which include the impact of quartz ampoules on He diffusion kinetics; concerns about possible sample heating during the sealing procedure; general suitability of the material for (U-Th)/He dating; and future improvements of the technique and addressed all comments of the reviewers.

All changes are tracked.

Best regards, Olga

---

## Referee Report (RR1)

**Referee Report**

**Manuscript ID:** egusphere-2024-992

**General comments:**

This is the second round of revisions. In the first round, the reviewers suggested several additional tests and requested more detailed information regarding the samples. However, these points have not been fully addressed in the revised manuscript. I understand that the authors are following the Associate Editor's suggestion to list the additional tests as future development. However, I strongly encourage the authors to conduct some of these tests prior to publication. Specifically, acquiring ages for washed goethite grains, heating grains to higher temperatures to ensure complete degassing, and measuring U and Th on unheated samples should be performed. Additionally, SEM images of the different types of goethite grains should be included in the supplementary information.

If the Associate Editor considers it sufficient to list the additional tests as future work, please disregard my request and proceed with publishing the manuscript following the implementation of minor revisions.

**Specific comments:**

Line 16: Replace "Fe- and Mn-oxide mineralization rocks" with "Fe- and Mn-oxide cemented rocks"

Line 20: My concern is that the authors generalize their observations from the fine-grained, mixed-phase, goethite-rich material to well-crystallized, massive goethite samples, which are commonly targeted in geochronology studies. Adsorbed U will be much less prominent on well-crystallized, massive goethite samples.

Line 20: Does *"multi-mineral grains"* mean "impure goethite grains"?

Lines 31-33: Farley et al (2024) show that goethite breaks down during vacuum step-heating experiments, precluding calculation of Helium diffusion parameters.

Line 59: Replace "fragments of Fe- and Mn-oxide mineralization" with "fragments of Fe- and Mn-oxide mineralized rocks"

Figure 2: Identify all peaks. Also, replace "Iron Oxide Hydroxides" labels with "Goethite", "Feroxyhyte", or "Ferrihydrite".

Lines 106-107: As far as can tell, there is no information on "standard material" in Flowers et al. (2022). The sample analyzed by Yakubovich et al. does not represent a standard material used in (U-Th)/He geochronology.

Lines 114, 117: Clarify the phrase "without extra cooling" – what does it mean?

Line 190: Replace "thus the quantification of the number of floating particles in the solution" with "thus determining the concentration of Mn and Fe in the solutions was not possible."

Line 191: From Table 3 - dark grain-1 (Th/U: 0.26, 0.06, and 0.32) and dark grain-2 (Th/U: 0.6, 1.3); vein grain-1 (Th/U: 1.4, 0.6, and 2.5) and vein grain-2 (Th/U: 1.7, 3.0). Interestingly, the observed Th/U ratios for grains that were not cleaned and those that were fall within similar ranges – dated yellow goethite grains Th/U: 2.4, 2.7 and yellow goethite grains used in the leaching experiment Th/U: 2.5, 3.0; dated dark grains Th/U: 0.3-0.8 and dark grains used in the leaching experiments Th/U: 0.3-1.3.

Line 287: *"The He loss from goethite is strongly controlled by radiation damage."* – What is the basis for this statement?

Lines 338-339: Delete the phrase: *"Alternatively, the small number of dated samples and distribution of samples may preclude being able to detect continuous Neogene mineralization throughout the region."*

Line 375-379: Reference appears in duplicate. Please correct it.

---

## Author Response (AR2)

Dear Marissa,

Thank you for the comments and suggestions. Below are point to point answer on the questions highlighted by you (blue color) and H.S. Monteiro (green color)

However, you will need to greatly expand section 5.1.3 Future Developments. Currently this section consists primarily of a short list of the additional tests/analyses that the original three reviewers requested. The manuscript needs to provide context for why doing these additional tests would be important. In other words, explain the major assumptions in your measurements that prompt these additional tests, and explain what carrying out these additional tests would mean for the methodology, considering different potential outcomes.

*Done. Lines 285-305*

Title: The revised title is still a bit confusing. I recommend the following revised title: "An analytical approach for (U-Th)/He dating of goethite by sample encapsulation in quartz ampoules under vacuum, with an example from the Amerasian Basin, Arctic Ocean

*Corrected*

Line 15: remove the word "new".

*Done*

Line 25: "mineral age" might be misleading. I read this as (U-Th)/He ages are crystallization ages, which in the vast majority of cases won't be true. Please reword.

*Agree. Removed this sentence.*

Lines 72-74: This new sentence is hard to follow, please consider rewording.

*Done*

Figure 2: Are the peaks associated with these other minerals (birnessite, quartz) shown? If so, please label them. What do the orange/yellow thin lines and downward arrows represent?

*We replaced the Figure with a new one where all peaks are marked. JFYI the lines and downward arrows represented the position for goethite peaks from the XRD database*

Lines 216-217: I do not understand what the authors mean by: "This coincidence might indicate that we are wrong when consider these measurements as erroneous." Please explain this more or try rewriting for clarity.

*Rephrased. "The coincidence of their (U-Th)/He ages might indicate that we were wrong when decided to exclude these grains from the consideration."*

Figure 5: Please plot the two ages and their uncertainties that are being excluded from the weighted mean. Since the authors are acknowledging that these ages may be correct, it is not justified to exclude them from the figure entirely.

*Added*

Line 244: Can you quantify what you mean by "large grains" here?

*Added. Large means visible, the crystal size > 50-70 mkm. In this case surface to volume ratio is low – surface U unlikely contribute more than first precents to the total U budget. Accurate calculation is a subject of a separate work as it should include crystallographic shapes, sorption capacity and U content by the grains.*

Lines 268-271: This point seems quite secondary to the list of additional tests that follow. Either move this text to the end of this section or delete it entirely. The argument being made here about detector type is a bit misleading too. For an example, an SEM is necessary for small samples/signal intensities. There are also much more appropriate citations for the reproducibility of Faraday cup measurements (it is usually much better than 1%).

*Removed this part. The value I presented for the Faraday cup was from the Helix SFT MS. Reproducibility was slightly more than 1%. I did not know that Faraday cup gives reproducibility much better than 1%. Thank you, we will check if there are any other factors that disturb the signal.*

Lines 272-284: The authors need to provide context for each of the additional tests that are listed. What are the limitations of the measurements the authors made, and the arguments that the reviewers brought up, that necessitate these additional tests?

*Done*

Line 307: What is "close modern eU content"?

*Rephrased. Close (similar) values of eU*

General comments:

... However, I strongly encourage the authors to conduct some of these tests prior to publication. Specifically, acquiring ages for washed goethite grains, heating grains to higher temperatures to ensure complete degassing, and measuring U and Th on unheated samples should be performed. Additionally, SEM images of the different types of goethite grains should be included in the supplementary information.
If the Associate Editor considers it sufficient to list the additional tests as future work, please disregard my request and proceed with publishing the manuscript following the implementation of minor revisions.

*We have added a reference to a fresh paper by Hein (accepted 24 of September) and coauthors where he describes DR7-001 sampling site and provide additional information on the rock and mineral geochemistry. As I understand one of the main concerns by HS Monteiro is that we used "nonstandard" goethite. So, we highlighted in the abstract and text that from the mineralogical point of view the goethite we used is a normal goethite, but it was formed at specific environment. Thus, it might be that the U is in a form that differs from its form in terrestrial supergene goethite. So, it is not clear if sonication would result in U-loss from the terrestrial samples.*

Specific comments:
Line 16: Replace "Fe- and Mn-oxide mineralization rocks" with "Fe- and Mn-oxide cemented rocks"

*Done*

Line 20: My concern is that the authors generalize their observations from the fine-grained, mixed-phase, goethite-rich material to well-crystallized, massive goethite samples, which are commonly targeted in geochronology studies. Adsorbed U will be much less prominent on well-crystallized, massive goethite samples.

*Rephrased. Tried to made it clear that we know nothing about U mobilization from terrestrial supergene goethite.*

Line 20: Does "multi-mineral grains" mean "impure goethite grains"?

*No, its multi-mineral grains. The mineral is an aggregate built from small crystals of goethite with some admixture of other minerals (quartz, Mn-oxide)*

Lines 31-33: Farley et al (2024) show that goethite breaks down during vacuum step-heating experiments, precluding calculation of Helium diffusion parameters.

*Thanks for the reference. Corrected.*

Line 59: Replace "fragments of Fe- and Mn-oxide mineralization" with "fragments of Fe- and Mn-oxide mineralized rocks"

*Done*

Figure 2: Identify all peaks. Also, replace "Iron Oxide Hydroxides" labels with "Goethite", "Feroxyhyte", or "Ferrihydrite".

*Replaced the figure for a new one with labels.*

Lines 106-107: As far as can tell, there is no information on "standard material" in Flowers et al. (2022). The sample analyzed by Yakubovich et al. does not represent a standard material used in (U-Th)/He geochronology.

*Rephrased. That goethite is a normal goethite, but the conditions of its formation is not usual*

Lines 114, 117: Clarify the phrase "without extra cooling" – what does it mean?

*When sample is sonicated in the ultrasonic bath it is heated. To avoid that, researches add ice to the water or use special temperature stabilizers. We did not provide extra cooling for the samples during the sonication. So actual temperature might be slightly higher than a room temperature.*

Line 190: Replace "thus the quantification of the number of floating particles in the solution" with "thus determining the concentration of Mn and Fe in the solutions was not possible."

*Done*

Line 191: From Table 3 - dark grain-1 (Th/U: 0.26, 0.06, and 0.32) and dark grain-2 (Th/U: 0.6, 1.3); vein grain-1 (Th/U: 1.4, 0.6, and 2.5) and vein grain-2 (Th/U: 1.7, 3.0). Interestingly, the observed Th/U ratios for grains that were not cleaned and those that were fall within similar ranges – dated yellow goethite grains Th/U: 2.4, 2.7 and yellow goethite grains used in the leaching experiment Th/U: 2.5, 3.0; dated dark grains Th/U: 0.3-0.8 and dark grains used in the leaching experiments Th/U: 0.3-1.3.

*Yes, the total values are almost equal, because the leaching resulted in loss of less than 5% of U. But the Th/U of leachates differs from the Th/U ratio of the analyzed grains.*

Line 287: "The He loss from goethite is strongly controlled by radiation damage." – What is the basis for this statement?

*Corrected the phrase based on the data provided by Farley et al., 2023. But originally this statement was on the basis of the article written by Bassal et al 2022 (numerical modeling)*

Lines 338-339: Delete the phrase: "Alternatively, the small number of dated samples and distribution of samples may preclude being able to detect continuous Neogene mineralization throughout the region."

*Deleted*

Line 375-379: Reference appears in duplicate. Please correct it.

*Corrected*